# Omnipresence of Partitiviruses in Rice Aggregate Sheath Spot Symptom-Associated Fungal Isolates from Paddies in Thailand

**DOI:** 10.3390/v13112269

**Published:** 2021-11-12

**Authors:** Sokty Neang, Santiti Bincader, Sansern Rangsuwan, Pisut Keawmanee, Soriya Rin, Lakha Salaipeth, Subha Das, Hideki Kondo, Nobuhiro Suzuki, Ikuo Sato, Daigo Takemoto, Chainarong Rattanakreetakul, Ratiya Pongpisutta, Masao Arakawa, Sotaro Chiba

**Affiliations:** 1Graduate School of Bioagricultural Sciences, Nagoya University, Nagoya 464-8601, Japan; neangsokty1@gmail.com (S.N.); rin.soriya@g.mbox.nagoya-u.ac.jp (S.R.); isato@agr.nagoya-u.ac.jp (I.S.); dtakemo@agr.nagoya-u.ac.jp (D.T.); 2Department of Plant Pathology, Faculty of Agriculture at Kamphaeng Saen, Kasetsart University, Nakhon Pathom 73140, Thailand; santiti.bi@ku.th (S.B.); sansern.ra@ku.th (S.R.); pisut.keaw@ku.th (P.K.); chainarong.r@ku.th (C.R.); agrryp@ku.ac.th (R.P.); 3Asian Satellite Campuses Institute–Cambodian Campus, Nagoya University, Phnom Penh 2696, Cambodia; 4School of Bioresources and Technology, King Mongkut’s University of Technology Thonburi, Bangkok 10140, Thailand; lakha.sal@kmutt.ac.th; 5Institute of Plant Sciences and Resources, Okayama University, Kurashiki 710-0046, Japan; subhadas1981@gmail.com (S.D.); hkondo@okayama-u.ac.jp (H.K.); nsuzuki@okayama-u.ac.jp (N.S.); 6Faculty of Agriculture, Meijo University, Nagoya 468-8502, Japan; aramasa@meijo-u.ac.jp

**Keywords:** rice aggregate sheath spot, mycovirus, Partitiviridae, Rhizoctonia oryzae-sativae, Rhizoctonia oryzae, protoplasting, virus-curing

## Abstract

Partitiviruses are one of the most prevalent double-stranded RNA viruses that have been identified mostly in filamentous fungi and plants. Partitiviruses generally infect host fungi asymptomatically but infrequently exert significant effect(s) on morphology and virulence, thus being considered a potential source of biological control agents against pathogenic fungi. In this study, we performed a screening for mycoviruses of a collection of Thai isolates of rice fungal pathogen *Rhizoctonia oryzae-sativae*, a causal agent of rice aggregated sheath spot disease. As a result, 36% of tested isolates carried potentially viral double-stranded RNAs with sizes ranging from 2 to 3 kbp. By conventional cDNA library construction and RNA-seq, we determined six new alphapartitiviruses that infected three isolates: tentatively named Rhizoctonia oryzae-sativae partitivirus 1 to 6 (RosPV1-6). Furthermore, RT-PCR detection of each virus revealed their omnipresent nature in different *R. oryzae-sativae* isolates. Although virus-curing of basidiomycetous fungi is generally difficult, our repeated attempts successfully obtained virus-free (for RosPV1, RosPV2, and uncharacterized partitiviruses), isogenic strain of *R. oryzae-sativae* TSS190442. The virus-cured strain showed slightly faster colony growth on the synthetic media and severe symptom development on the rice sheath compared to its virus-infected counterpart. Overall, this study shed light on the distribution of partitiviruses in *R. oryzae-sativae* in a paddy environment and exemplified a virus-curing protocol that may be applicable for other basidiomycetous fungi.

## 1. Introduction

Rice (*Oryza sativa* L.) is a staple food crop in most Asian countries, and its production is threatened by many pathogenic microbes, such as *Rhizoctonia solani* (rice sheath blight), *Pyricularia oryzae* (leaf blast), and *Xanthomonas oryzae* pv. *oryzae* (bacterial blight) [1]. In addition, *R. oryzae* and *R. oryzae-sativae* (teleomorph: *Waitea circinata*, *Ceratobasidium oryzae-sativae*, respectively) are soil-borne basidiomycetous fungal pathogens known to cause aggregate sheath spots on rice plants worldwide [2,3]. Both pathogens withstand hostile environments. Aggregated sheath spots diseases are reported to cause yield loss up to 20.3% in Australia and 9% in Uruguay; also, such diseases were reported as a significant threat to rice production in California, the USA [2,4].

Mycoviruses, viruses that infect fungi, have attracted much attention from researchers with their potential as a biocontrol (virocontrol) agent against phytopathogenic fungi [5,6]. For example, the application of Cryphonectria hypovirus 1 to control the chestnut blight fungus *Cryphonectria parasitica* [7,8] has led to discoveries of several mycoviruses with biocontrol potentials, such as Sclerotinia sclerotiorum hypovirulent-associated DNA virus 1 and Sclerotinia sclerotiorum partitivirus 1 (SsPV1) of the white mold fungus, Rosellinia necatrix megabirnavirus 1 of the white root rot fungus, and others [9,10,11]. Likewise, the rice blast fungus *P. oryzae* (formerly *Magnaporthe oryzae*) is reported to carry several mycoviruses, and, of those, Magnaporthe oryzae chrysovirus 1-A and 1-B are known to carry cytotoxic protein genes [12,13]. For instance, of mycoviruses in rice-infecting *Rhizoctonia* species, a capsidless positive-sense (+) single-stranded RNA (ssRNA) mycovirus, Rhizoctonia solani endornavirus 1 (RsEV1), was previously reported to confer hypovirulence to *R. solani* AG-1-IA GD 118-P, a causal agent of rice sheath blight disease. In addition, RsEV1 infection significantly downregulated the synthesis of serine and tyrosine, amino acids (aa) that are involved in the biosynthesis of secondary metabolites in its host fungus [14]. 

To the best of our knowledge, only two mycoviruses infecting *R. oryzae-sativae* are reported to date. Rhizoctonia oryzae-sativae mitovirus 1 (RoMV1) was detected in *R. oryzae-sativae* strain 89-1 isolated from rice sheath showing typical symptoms of aggregate sheath spot in China [15]; Rhizoctonia oryzae-sativae partitivirus 1 (RosPV1) was characterized from a rice-infecting field isolate of *R. oryzae-sativae* (strain RS005) in India [16]. However, the influence of these mycoviruses on host pathogenicity has not been investigated.

*Partitiviridae* is an expanding double-stranded RNA (dsRNA) virus family that currently consists of five established genera (*Alpha*-, *Beta*-, *Gamma*-, *Deltapartitivirus,* and *Cryspovirus*) [17], and it may accommodate more diverse groups of related viruses, i.e., proposed genera tentatively named as “Zetapartitivirus” and “Epsilonpartitivirus” [10,18,19,20]. Members or potential members of *Partitiviridae* are infectious to plants, fungi, protozoa, and insects, in which fungal partitiviruses (approved members) are frequently detected in ascomycetous and basidiomycetous fungi and classified in the genera *Alpha*-, *Beta*-, and *Gammapartitivirus*. Limited numbers of partitiviruses are reported to confer hypovirulence to host fungi that include Rosellinia necatrix partitivirus 10 in association with Rosellinia necatrix virga-like virus, Botrytis cinerea partitivirus 2, SsPV1, and Rhizoctonia solani partitivirus 2 (RsPV2) [17].

This study attempted to isolate rice sheath bright fungus and aggregate sheath spot fungus from paddy fields in Thailand and detected several mycovirus-associated dsRNA elements in a total of 21.3% of isolates screened. Here, we report the genomic and phylogenetic characteristics of those mycoviruses and their distribution in the respective fungal population. Moreover, the biological impact of selected viruses is shown via a newly developed virus-curing protocol.

## 2. Materials and Methods

### 2.1. Fungal Collection, Culture, and Maintenance

A total of 80 fungal isolates were isolated from 200 symptomatic rice plants from paddy fields in Don Chedi and Song Phi Nong districts of Suphan Buri province, Thailand in March 2019. All the collected rice tissues were surface sterilized with 1.2% sodium hypochlorite (NaOCl), washed twice with ddH_2_O, then placed on 2% water agar (WA) containing 50 μg/mL streptomycin followed by 24 h incubation at 25 °C. The tip of newly elongated hyphae was cut and transferred to potato dextrose agar (PDA) plates for single specimen isolation and maintained at room temperature for regular use. For long-term or short-term storage, all isolates were cultured on sterilized glass fiber (Advantec, Tokyo, Japan), dried then stored at −30 °C, respectively. Based on the colony morphology of those isolates, 33 *Bipolaris*-like isolates and 47 *Rhizoctonia*-like isolates were distinguished. 

### 2.2. Taxonomic Analysis of Fungal Species

Ten of approximately 25 mm^2^ mycelium plugs were used to inoculate Potato Dextrose Broth (PDB) and culture flasks were incubated at 25 °C in the dark for 5 days. The mycelial mats were harvested, and then homogenized into a fine powder in the presence of liquid nitrogen. The pulverized tissues were subjected to genomic DNA extraction using DNeasy Plant Mini-Kit following the manufacturer’s instructions (Qiagen, Venlo, Netherland). For taxonomic classification of fungal isolates, the internal transcribed spacer (ITS) regions were then amplified in a polymerase chain reaction (PCR) using the universal primer set, ITS4 (5′-TCCTCCGCTTATTGATATGC-3′) and ITS5 (5′-GGAAGTAAAAGTCGTAACAAGG-3′). The PCR products were electrophoresed on 1% (*w*/*v*) agarose gel and visualized by ethidium bromide staining using a UV transilluminator. Subsequently, PCR products were purified using a Gel Clean-up Kit (Wizard SV Gel and PCR Clean-Up System, Promega, Madison, WI, USA) and quantified using a BioSpec-Nano (Shimadzu, Tokyo, Japan). PCR products were then subjected to Sanger Sequencing in both the forward and reverse directions using an Applied Biosystems 3130 Genetic Analyzer.

### 2.3. dsRNA Extraction

Total dsRNA fractions were extracted from 5 mg of mycelia by conventional cellulose affinity column chromatography, as previously described [21]. Briefly, fungal mycelial mats cultured in PDB were filtrated and homogenized using a mortar and pestle in the presence of liquid nitrogen. Total RNA fractions were obtained with a series of centrifugation using phenol-chloroform-isoamyl alcohol (PCI) 25:24:1 (*v*/*v*/*v*) and chloroform-isoamyl alcohol (CIA) 24:1 (*v*/*v*). dsRNA fractions were then purified by another round of centrifugation using a mixture of 1× Sodium/Tris/EDTA (100 mM NaCl, 10 mM Tris-HCl pH 8.0, and 1 mM EDTA pH 8.0) containing 16% ethanol and Cellulose Powder C (Advantech). The purified solution was further treated with S1 Nuclease and RQ1 RNase-Free DNase I (Promega) to eliminate genomic DNA and single-stranded RNA (ssRNA) contaminants. The presence of viral dsRNAs was confirmed by 1% agarose gel electrophoresis in 0.5× Tris/Borate/EDTA (40 mM Tris, 45 mM boric acid, and 1 mM EDTA).

### 2.4. cDNA Library Construction, RT-PCR, and RLM-RACE

The purified dsRNAs were used to construct complementary DNA (cDNA) libraries. dsRNA fragments were pre-denatured at 99 °C for 5 min, and then subjected to first-strand cDNA synthesis by reverse transcription using ReverTra Ace (TOYOBO, Tokyo, Japan) with an adapter-tagged random hexamer primer (5′-CCTGAATTCGGATCCTCCNNNNNN-3′). The second-strand synthesis was carried out using a single primer (5′-CCTGAATTCGGATCCTCC-3′), in which its sequence matches that of the adapter primer, and GoTaq Green Master Mix polymerase (Promega). DNA fragments were then separated by 1% (*w*/*v*) agarose gel electrophoresis (by running the gel at 100 V for 40 min), and DNA amplicons ranging between 1.0 and 2.0 kbp were exclusively purified using the Gel Clean-up Kit. The purified DNA fragments were subsequently ligated into pGEM T-easy vector (Promega) and the ligation products were used for transforming *Escherichia coli* DH5α competent cells. Recombinant plasmids were extracted and sequenced at both directions using universal M13F and M13R primers. The remaining first-strand cDNA products were used to fill the gaps between resulting contig sequences using gene-specific primers (respective primer sequences have been provided in Appendix A). 

To obtain 5′ and 3′ terminal sequences, RNA ligase-mediated rapid amplification of cDNA ends (RLM-RACE) was performed. dsRNAs were initially denatured in 90% dimethyl sulphoxide (DMSO) at 65 °C for 20 min. Subsequently, 5′ phosphorylated DNA oligonucleotide adaptor (5′-PO4-CAATACCTTCTGACCATGCAGTGACAGTCAGCATG-3′) was ligated to 3′ termini of both dsRNA strands using T4 RNA ligase (TaKaRa, Kusatsu, Japan) at 16 °C for 16–18 h. Next, DMSO was added to the DNA adaptor-ligated dsRNA to a final concentration of 90% in the presence of oligonucleotide 3′-RACE-1st (5′-CATGCTGACTGTCACTGCAT-3′), which binds to the last 20 nt of the ligated adaptor sequence. The freshly denatured adaptor-ligated dsRNAs were used as templates for cDNA synthesis using M-MLV Reverse Transcriptase (Invitrogen, Thermo Fisher, Waltham, MA, USA). The synthesized cDNAs were amplified by PCR using 3′-RACE-2nd (5′-TGCATGGTCAGAAGGTATTG-3′) and respective gene-specific primers (Appendix A) targeting 5′ and 3′ termini. RACE amplicons were cloned and sequenced by the Sanger method. The sequence for each amplicon was determined by sequencing at least three independent clones.

### 2.5. Next-Generation Sequencing (NGS) Approach

Total RNA fraction was extracted using RNAiso Plus kit (TaKaRa) following the manufacturer’s protocol. The total RNA samples from *R. oryzae-sativae* TSD190123, *Nigrospora oryzae* TSD190144, and *R. solani* (Japanese isolate) were pooled together (3.3 µg/µL, RIN = 9.1) and characterized by NGS (RNA-seq) analysis. The ribosomal RNA (rRNA)-depleted samples were then employed as a template for the construction of cDNA libraries and pair-end sequencing using the Illumina HiSeq 2000 platform (Illumina, Hayward, CA, USA). The rRNA depletion, cDNA library construction, and deep sequencing analysis were performed by Macrogen Japan (Kyoto, Japan). Subsequently, adapter sequences were removed, and trimmed high-quality clean reads (62.5 M reads) were assembled into individual contigs (total 43998) using the CLC Genomic Workbench ver. 8 (CLC bio, Aarhus, Denmark). In this study, partitivirus-like contigs were squeezed and the presence of those sequences in TSD190123 were confirmed by RT-PCR. To obtain 5′ and 3′ terminal sequences, 3′ RLM-RACE was performed following the aforementioned methods (primer sequences are provided in Appendix A). Since we focused on partitiviruses in this study, the remaining viral sequences (contigs) from TSD190123 and the other fungal isolates will be reported elsewhere (Neang et al., unpublished data).

### 2.6. Bioinformatic Analysis

The vector contaminations and RACE adaptor sequences were removed from all the obtained sequences of cDNA clones using an online tool, VecScreen (https://www.ncbi.nlm.nih.gov/tools/vecscreen/, accessed on 2 November 2021). The resulting clean sequences were then assembled (with at least 80% identity) into contigs using ATSQ software (Genetyx Inc., Tokyo, Japan). The Open Reading Frames (ORF) Finder program (https://www.ncbi.nlm.nih.gov/orffinder/, accessed on 2 November 2021) was applied to determine the gene coding regions in viral contigs. Deduced aa sequences of coding proteins, RNA-dependent RNA polymerase (RdRp), and coat protein (CP), were predicted using Genetyx software (Genetyx Inc., Tokyo, Japan). The identification of virus-like sequences and their genetic relatedness to other viral sequences was analyzed with the online Basic Local Alignment Search Tool (BLAST). Subsequently, previously reported mycovirus sequences were downloaded from the National Center for Biotechnology Information (https://blast.ncbi.nlm.nih.gov/, accessed on 2 November 2021). The full-length aa sequences were subjected to phylogenetic analyses. These aa sequences were aligned using the online MAFFT program (https://mafft.cbrc.jp/alignment/server/index.html, accessed on 3 November 2021). Maximum-likelihood (ML) phylogenetic trees were constructed based on the aligned aa sequences using PhyML with the automatic model selection mode (http://www.atgc-montpellier.fr/phyml/, accessed on 3 November 2021). The ML trees were visualized and polished in FigTree v1.4.4 (http://tree.bio.ed.ac.uk/software/figtree/, accessed on 3 November 2021).

### 2.7. Virus Curing

To eliminate mycoviruses from *R. oryzae-sativae* TSS190442, protoplast purification, hyphal tip isolation, post-heat, and anti-viral drug treatments were applied. All incubation steps were carried out at 25 °C unless otherwise indicated.

Protoplasts were obtained by incubation of 16-hr-old mycelia in 0.6 M MgSO_4_, pH 5.2 containing Driselase (20 mg/mL) (Sigma-Aldrich, St. Louis, MO, USA), Cellulase RS (20 mg/mL) (Yakult Pharmaceutical Ind. Co., Tokyo, Japan), Lysozyme (10 mg/mL) (Fujifilm Wako Pure Chemical Corporation, Osaka, Japan), and Lysing enzyme (10 mg/mL) (Sigma-Aldrich, St. Louis, MO, USA) for 3 h at 35 °C with 55 rpm aerial shaking. Protoplasts were purified by placing trapping buffer (0.3 M sorbitol in 100 mM Trish-HCl pH 7.0) on top of the enzymatic digestion solution in a ratio of 1:1. Subsequently, protoplasts were collected from the interphase. Healthy protoplasts were then recovered and grown on a regeneration medium (RM) (1 g/L yeast extract, 1 g/L casein hydrolysate, 1 M sucrose, and 1.5% agar). Hyphal tips from germinated mycelia were transferred onto new RM plates and screened for the virus infections by one-step colony RT-PCR with virus-specific primers (Appendix A).

The recovered fungal derivatives were further incubated in the dark at 37 °C for 5 days. Hyphal tips were later transferred onto new PDA plates and cultured in the dark at room temperature for another 5 days before being subjected to secondary mycovirus screening by one-step colony RT-PCR. 

The mycelial plugs of isolate TSS190442 were cultured on the edge of PDA plates supplemented with ribavirin (0.2 mg/L) with or without cycloheximide (5 mg/L) for 3 days Subcultures were obtained from the tip of actively growing mycelium and transferred to fresh media. This step was repeated five times. Finally, sixth subcultures were placed and incubated on drug-free PDA. Newly germinated mycelia were subjected to one-step colony RT-PCR to screen for mycovirus infection. For further confirmation, total dsRNAs were extracted and profiled from detection-negative fungal colonies (grown in PDB) following the methods described previously. 

### 2.8. Detection of Partitiviruses by RT-PCR

Detection of partitivirus infection among all the fungal isolates was conducted by RT-PCR using partitivirus-specific primers (Appendix A) that were designed from the RdRp-coding regions from partitivirus genome sequences identified in this study. Briefly, a portion of supernatant from the CIA step in dsRNA extraction (see above) was precipitated with lithium chloride (2 M final concentration) to enrich the ssRNA fraction. The obtained ssRNAs were further treated with RQ1 RNase-Free DNase I (Promega) before being subjected to cDNA synthesis. Subsequently, the cDNA was used as a template in PCR reaction along with gene-specific primers targeting individual partitiviruses. The presence of partitivirus was confirmed by 1.0% agarose gel electrophoresis of RT-PCR products.

### 2.9. Pathogenicity Test

To investigate the impact of mycoviruses on host fungal growth and virulence, the sizes of fungal colonies on PDA and symptoms on plants were elucidated. The mycelial growth of virus-infected *R. oryzae-sativae* TSS190442 and its virus-cured derivative on PDA were analyzed by measuring the radius of colonies for 7 days. Standard deviation and T-test were calculated on five replicates for each isolate.

The virulence of *R. oryzae-sativae* isolates was evaluated by an in vitro assay that is recently developed for *R. solani* pathogenicity (Arakawa et al., unpublished). Two-month-old rice (cv. Koshihikari) stems were detached from slightly above the soil, sterilized the surface with 1.2% NaOCl, then immediately transferred into a sterilized 14 cm test tube containing 3 mL distilled water. A piece of PDA block with actively growing mycelia of virus-free and virus-infected *R. oryzae-sativae* isolates were then placed on the node section between rice stem and first/second leaf. Excess rice tissues 3 cm above the inoculation site were chopped off. All tubes were sealed with sterilized silicon caps and maintained at 25 °C with 12 h light/dark condition. The degree of pathogenicity was evaluated by measuring the length of symptomatic area (spotted blight lesions) on the sheath from the inoculation site downwards. Standard deviation and T-test were calculated on three replicates of each isolate.

## 3. Results

### 3.1. Determination of Fungal Species 

On water–agar plates, hyphal tip cultures of fungal mycelia that were originally developed from rice tissues showing blight-like symptoms (Figure 1A), exhibited *Rhizoctonia*-like and *Bipolaris*-like colony morphologies (Figure 1B,C). One of the isolates, TSD190117, showed strongly impaired growth (Figure 1D). Genomic DNA of 80 fungal isolates was extracted and used as a template for amplification of the ITS regions (540 to 849 bp) and sequencing. The BLASTn search with amplicon sequences revealed the best hit with ITS sequences of *Rhizoctonia* species (42 isolates), *Bipolaris* species (8 isolates), *Fusarium* species (7 isolates), *Nigrospora* species (6 isolates), *Gaeumannomyces* species (5 isolates), and others (12 isolates) (Table 1). A total of 13 out of 42 isolates related to the *Rhizoctonia* species showed 100% identity among each other. Detailed information about the fungal collection is summarized in Appendix A.

### 3.2. Screening of Fungal Isolates for Viral dsRNAs

The dsRNA elements were isolated and profiled from 80 fungal isolates derived from rice sheaths showing blight-like symptoms. Of the 80 isolates, 17 isolates (21.3% of tested fungal isolates) were found to carry dsRNA elements, which was further confirmed based on their resistance to double digestion by S1 Nuclease and DNase I (Table 1 and Figure 1E). Agarose gel electrophoretic profiles showed commonly the presence of multiple dsRNA elements ranging from 1.5 to 2.0 kbp in length in all *R. oryzae-sativae* and *R. oryzae* isolates (Figure 1E). These dsRNAs could be the genomes of partitiviruses, curvulaviruses, and others based on the range of dsRNA sizes. *R. solani* TSD190117 with strongly impaired growth carried similar dsRNA elements (Appendix A), but the fungal isolate was no longer viable in the successive culture. In addition, *Nigrospora oryzae* TSD190144 and *N. sphaerica* TSD190109 appeared to carry a ≥10 kbp dsRNA, and 2.5 and 6 kbp dsRNAs, respectively (Figure 1E). Likewise, *R. oryzae* TSS190517 and *R. zeae* TSS190513 harbored large dsRNA elements. Given that the dsRNA banding patterns from different *R. oryzae-sativae* isolates were similar to each other (Figure 1E, green letters), dsRNAs in isolates TSS190442 and TSS190505 were chosen for further molecular characterization. 

### 3.3. Genomic Organization of Analyzed dsRNA Elements

Sanger sequencing of cDNA clones and RACE clones of dsRNAs in *R. oryzae-sativae* isolates TSS190442 and TSS190505 revealed the complete genomes of three partitiviruses (six dsRNA segments; see below). Following the nomenclature guidelines of ICTV, these fully-sequenced partitiviruses were named Rhizoctonia oryzae-sativae partitivirus 1 to 3 (RosPV1, 2, and 3). Note that one of the viruses in TSS190442, RosPV1, is handled as a different strain of RosPV1-RS001 [16]. Furthermore, NGS results detected three additional partitiviruses (total six dsRNA segments) infecting *R. oryzae-sativae* TSD190123, designated as Rhizoctonia oryzae-sativae partitivirus 4 to 6 (RosPV3, 4, and 6). The assignment of two genomic segments to one virus was based on their strictly conserved 5′-terminal end sequences (see below). The complete genome sequences of all the six viruses were submitted to GenBank/DDBJ/ENA with accession numbers: LC637850, LC637851, LC637852, LC637853, LC637854, LC637855, LC637856, LC637857, LC637858, LC637859, LC637860, and LC637861.

The 12 dsRNA segments were separated into two genomic groups. 1) larger dsRNAs (dsRNA1) of RosPV1 to RosPV6: these are 1951, 1995, 1988, 2004, 2008, and 1897 bp in length with GC content 43%, 44.1%, 46.2%, 47%, 43.4%, and 46.2%, respectively. Each dsRNA contains one ORF hypothetically encoding an RdRp with a predicted molecular weight of 69.5, 73.2, 72.7, 72.6, 72.5, and 68.7 kDa. 2) shorter dsRNAs (dsRNA2) of RosPV1 to RosPV6: these are 1808, 1775, 1815, 1768, 1914, and 1852 bp in length with 44.5, 48.3, 47.2, 47.6, 47.8, and 50.1% GC content, respectively. One ORF was predicted in each segment, encoding for a putative CP with an estimated molecular mass of 54.1, 53.6, 55.1, 54, 63.8, and 58.4 kDa (Figure 2A–C).

Untranslated regions (UTRs) of these viral genome segments were generally short, as 5′-UTRs ranged from 65 to 111 bp, and those of 3′-UTRs were a little varied but their sizes ranging between 54 and 265 bp including poly(A) stretch (Figure 2C). The 5′ termini in the given dsRNA set of RdRp-coding dsRNA1 and CP-coding dsRNA2 segments were well conserved. Whereas, the 3′ terminal regions were moderately similar. Of note, five 3′-ends of CP-coding dsRNA2 segments, except for RosPV4, contained poly(A) tail, but all six of RdRp-coding segments appeared to have integrated poly(A). The numbers of A residues in these terminal sequences varied among RACE clones; thus, the final sequences presented here are based on the consensus (sequences that occurred most frequently). The 5′ termini (approximately 23–24 nt) of the 12 dsRNA segments were aligned, and the dsRNA pair, which constitutes the single viral individual, possessed identical nucleotides in the initial 7 to 17 nt (Figure 2D). In the alignment with other alphapartitiviruses, a conserved cluster at the 5′-ends (GAA) was identified (except for RosPV2, which has 5′-GUA), suggesting that these viruses may be members of the genus *Alphapartitivirus* [22,23] (Figure 2D, highlighted in pink). 

### 3.4. BLAST Search with Deduced Amino Acid Sequences of Viral Proteins

BLASTp analyses based on deduced aa sequences revealed that the 12 dsRNAs were the six genomic segment sets of RosPV1 to RosPV6. The predicted aa sequences of RdRp and CP encoded by all six RosPVs had the highest hits with the following previously reported partitiviral RdRps and CPs:

RosPV1-T442, RosPV1-RS005 discovered in India (accession number MK015642 and MK015643) [16] (RdRp, E-value: 0.0; Query cover: 90%; Identity: 93.3%) (CP, E-value: 0.0; Query cover: 100%; Identity: 91.3%).

RosPV2-T442, Rhizoctonia solani partitivirus 4 (RsPV4) (accession number KX914902 and KX914903) [24] (RdRp, E-value: 0.0; Query cover: 90%; Identity: 67.3%) (CP, E-value: 1×10^−110^; Query cover: 88%; Identity: 42.8%).

RosPV3-T505, unreported virus sequences deposited in 2015 as Rhizoctonia fumigata partitivirus (RfPV) C314 strain (accession number KM668042 and KM668043) (RdRp, E-value: 0.0; Query cover: 91%; Identity: 99.4%) (CP, E-value: 0.0; Query cover: 100%; Identity: 94.5%).

RosPV4-T123, Rhizoctonia solani dsRNA 4 (accession number YP_009551520 and YP_009551521) (RdRp, E-value: 0.0; Query cover: 100%; Identity: 84.1%) (CP, E-value: 1×10^−161^; Query cover: 99%; Identity: 50.1%).

RosPV5-T123, Rhizoctonia solani partitivirus 7 (RsPV7) (QDW81313 and QDW81314) [25] (RdRp, E-value: 0.0; Query cover: 99%; Identity: 67.1%) (CP, E-value: 3×10^−114^; Query cover: 94%; Identity: 39.3%).

RosPV6-T123, Rhizoctonia solani partitivirus 5 (RsPV5) (AZQ25369 and AZQ25369) [26] (RdRp, E-value: 0.0; Query cover: 100%; Identity: 83.7%) (CP, E-value: 2×60^−110^; Query cover: 92%; Identity: 32.7%).

### 3.5. Phylogenetic Analysis of Partitiviruses 

Complete aa sequences of RdRps and CPs were subjected to the construction of a multiple sequence alignment and the subsequent phylogenetic analysis (Figure 3). The six partitiviruses appeared to have a close relationship with the previously reported alphapartitiviruses (grouped in the same clade). As phylogenetic trees constructed based on the RdRp and CP aa sequences showed similar topology, these suggested that the pre-speculated combinations of RdRp/CP-coding segments as a virus genome were all reasonable (Figure 3A,B). As expected from BLAST analyses, RosPV1-T442 and RosPV3-T505 were phylogenetically very closely related to RosPV1-RS005 and RfPV, respectively. In contrast, RosPV2-T442, RosPV4-T123, RosPV5-T123, and RosPV6-T123 were substantially distant from reported partitiviruses (Figure 3A,B). Altogether, we propose that RosPV2-T442, RosPV4-T123, RosPV5-T123, and RosPV6-T123 represent novel alphapartitivirus species to be established, while RosPV1-T442 and RosPV3-T505 belong to the same species that accommodates RosPV1-RS005 and RfPV-C314, respectively.

### 3.6. RT-PCR Based Detection of Partitiviruses from Fungal Isolates Used in This Study

*Rhizoctonia* isolates collected in this study showed similar dsRNA profiles, and this made us speculate the multiple coinfections of the six characterized partitiviruses. Therefore, the detection of RosPV1 to RosPV6 in dsRNA-positive isolates was performed by RT-PCR with specific primers (Appendix A). In 16 fungal isolates tested, these partitiviruses were detected in eight isolates of *R. oryzae-sativae* (Figure 4). However, any of these partitiviruses were not found in *N. oryzae* TSD190144, *N. sphaerica* TSD190109, *R. oryzae* TSS190517, and *R. zeae* TSS190513 that accommodated dsRNA elements of over 3 kbp (Figure 1). The most prevalent RosPV2 was found in six *R. oryzae-sativae* isolates, and RosPV6 was similarly detected in five *R. oryzae-sativae* isolates. RosPV3 was detected from four *R. oryzae-sativae* isolates, while RosPV1 and RosPV5 were present in three *R. oryzae-sativae* isolates. RosPV4-T123 was detected only in two *R. oryzae-sativae* isolates (Figure 4). These results suggested that most of the six partitiviruses were widespread in *R. oryzae-sativae* population with the ability to coinfect a single isolate. For example, *R. oryzae-sativae* TSD190113 was coinfected by RosPV2-, RosPV3-, RosPV6-, and RosPV1-like viruses. Interestingly, none of the six partitiviruses were detected in the four *R. oryzae-sativae* isolates (TSD190106, TSD190119, TSD190136, and TSD190144) despite the fact that dsRNA profiles for these four isolates were similar to those of partitivirus-infected isolates (Figure 1 and Figure 4). It is possible that these isolates could harbor either partitiviruses unrelated to those of detected in this study or other small dsRNA or ssRNA viruses. From the different PCR band intensities (Figure 4), the accumulation levels of each mycovirus may vary among *R. oryzae-sativae* isolates, and/or genetic variation of partitiviruses in each viral strain might be rich, but these need further investigations to conclude.

### 3.7. Other Partitiviruses Coinfecting R. oryzae and R. oryzae-sativae

During the characterization of RosPV1–3, we have also detected sequence contigs that closely related to but different from these viral sequences, indicating the presence of additional mycoviruses in *R. oryzae-sativae* TSS190442 and TSS190505. Briefly, the BLASTx analysis of these independent cDNA contigs suggested coinfection by distinct partitivirus(es) sharing aa sequence similarity (52–92%) to RosPV3, RsPV2, Rosellinia necatrix partitivirus 6 (RnPV6), and Brassica rapa cryptic virus 1 (BrCV1) in TSS190442; and those similar to RosPV2 (94%) and RsPV3 (15%) in TSS190505 (Table 2). In addition, partially characterized cDNA libraries from *R. oryzae-sativae* isolates TSD190103, TSD190108, and TSS190401 revealed sequence contigs sharing the deduced aa sequence identities 37–99% with a member of *Alphapartitivirus*. These results are reasonable if compared to the RT-PCR detection pattern in Figure 4, except the fact that RosPV3 in TSD190108 and TSS190401 could not be detectable by RT-PCR but the similar contigs of RosPV3 could be detected in sequencing. This could be due to mismatches between primers and targets. Taken together, this study, for the first time, revealed the omnipresence nature of diverse partitiviruses in *R. oryzae* and *R. oryzae-sativae* infecting rice in Thailand.

### 3.8. Isogenic Virus-Cured Isolates

In general, virus-curing in basidiomycete fungi including *Rhizoctonia* spp. is difficult, and several attempts were reported to be unsuccessful (Table 3). However, it is reported that a hypovirus was successfully eliminated from the *R. solani* XN84 [27]. In this study, we tested previously reported virus-curing methods, such as heat treatment, protoplasting, and drug treatment in conjunction with hyphal tipping using the randomly selected *R. oryzae-sativae* TSS190442. After 5 days incubation at 37 ℃ (heat treatment), the fungal growth was severely hindered. However, its sub-cultures could reclaim a normal growth rate when incubated at room temperature. Unfortunately, no virus-free strain were obtained following this method.

All the regenerated fungal colonies derived from TSS190442 protoplasts were found to be positive for the presence of RosPV1 and RosPV2 in an RT-PCR-based detection. Hyphal tipping in conjunction with the treatment of anti-viral drug ribavirin is often useful in virus-curing. However, this method also failed to eliminate viruses from TSS190442 in this study (Table 3).

Finally, isogenic virus-free *R. oryzae-sativae* isolates were successfully obtained from TSS190442 through repetitive hyphal tipping from a culture growing on PDA supplemented with ribavirin (200 μm/L) combined with cycloheximide (5 mg/L) (Table 3). It is noteworthy that the use of cycloheximide coupled with ribavirin was essential to eliminate partitiviruses from *R. oryzae-sativae* TSS190442, especially when hyphal tipping from regenerated protoplasts and heat-treatment completely failed viral elimination. To examine the effect of partitivirus infection on the morphology and phenotype of *R. oryzae-sativae*, a representative virus-free sub-isolate T442-VF10 was selected for growth rate and colony morphology assays along with the virus-infected isolate (Figure 5A). The elimination of RosPV1 and RosPV2 in T442-VF10 was confirmed several times by one-step colony RT-PCR, conventional RT-PCR (using purified ssRNA fractions as template), and dsRNA extraction (Figure 5B; Appendix A). In addition, no presence of partially characterized partitiviruses (RosPV3-like, RsPV2-like, and RnPV6-like viruses) in T442-VF10 was confirmed as well. With the absence of RosPV1, RosPV2, and other viruses, the growth rate of T442-VF10 on PDA was slightly improved, and it appeared to have developed denser mycelia and more aerial hyphae compared to its virus-infected counterpart (TSS190442) (Figure 5A,C). The pathogenicity of the two variants was tested on detached rice sheaths. The result showed that the virus-free isolate induced larger lesions (Figure 5D, see the lesion scale bars), albeit with no statistical significance (Figure 5E).

## 4. Discussion

### 4.1. Multiple Infections of Partitiviruses in Rhizoctonia *spp.*

A total of 21% of all tested fungal isolates carry multiple dsRNA elements with a potential of viral origin (Figure 1, Table 2 and Table 3), although dsRNA-negative isolates might also carry mycoviruses that were below the detection limit. Six dsRNA sets among all were characterized, RosPV1- and RosPV2-T442 infecting *R. oryzae-sativae* TSS190442, RosPV3-T505 infecting TSS190505, and RosPV4-, RosPV5-, and RosPV6-T123 infecting TSD190123. The six mycoviruses were identified as alphapartitiviruses in the family *Partitiviridae* based on BLASTp search results and phylogenetic study (Figure 3). Although the ITS sequence alone is not sufficient to accurately confirm the fungal taxonomic placement, it showed that *R. oryzae-sativae* TSD190106, TSD190113, TSD190119, and TSD190133 were considered genetically close (100% identical ITS sequences) despite harboring marginally different dsRNA profiles (Figure 1E, Appendix A). Likewise, *R. oryzae-sativae* TSD190103 and TSD190108 were indistinguishable based on their ITS sequences. We assumed that dsRNA elements detected in these fungi include partitiviruses since their dsRNA pattern is somewhat similar to partitiviruses identified in this study. In addition, these *R. oryzae-sativae* isolates are potentially infected by non-partitiviruses but similar-sized RNA viruses; for instance, the isolate TSD190123 subjected to RNA-seq was found to be infected at least eight RNA mycoviruses, excluding RosPV4, 5, and 6 (data not shown). Further, this study revealed that RosPV1-T442, RosPV2-T442, RosPV3-T505, and RosPV6-T123 together coinfect *R. oryzae-sativae* TSD190113 (Figure 4). The coinfection of a single fungal strain by multiple partitiviruses is interesting but not unprecedented as coinfection by multiple partitiviruses of the same host fungus has been reported in *Rhizoctonia* spp. For instance, two alphapartitiviruses, RsPV3 and RsPV4, coinfected *R. solani* strain HG81 isolated in Hubei province, China [24], three betapartitiviruses, RsPV6–8, were discovered from *R. solani* isolate YNBB-111 [31], and a Brazilian isolate of *R. solani* (strain IBRS23) was also reported to harbor one betapartitivirus and two alphapartitiviruses, RsPV6–8 [25]. It is worth noting that RsPV6 characterized in China is 100% identical to RsPV6 from Brazil, while the RsPV7 and eight discovered from both regions were distinct partitiviruses. This phenomenon is more common than it was previously thought, and found to occur not only in *Rhizoctonia* spp., but also in *Rosellinia necatrix* [23]. The possibility of interactions between the coinfecting partitiviruses even from different genera (*Alphapartitivirus* and *Betapartitivirus*) is being explored.

### 4.2. Widespread Nature of the Partitiviruses Found in This Study

Electrophoretic dsRNA profiles and sequencing analysis indicated the widespread nature of partitivirus infection. Out of 17 dsRNA-positive *R. oryzae-sativae* isolates, 7 were infected by multiple partitiviruses (Figure 1E and Figure 5). According to previous studies, viruses in the family *Partitiviridae* can be efficiently transmitted by horizontal (via hyphal anastomosis) and vertical transmission (via conidiospore or basidiospore) [10,22,32,33]. Indeed, *Rhizoctonia* spp. is widespread globally by the relocation of sclerotia and fragments of mycelium through infected plant samples, rarely by dispersion of basidiospore, and by a natural force such as wind and rain [34,35]. This nature of *Rhizoctonia* spp. and the high transmission rate of partitivirus may contribute to the prevalence of partitiviruses as reported in this study. For instance, RosPV3-T505 infecting *R. oryzae-sativae* TSD190505 isolated from a paddy field in Thailand was almost identical in genomic sequence to RfPV detected in China; similarly, RosPV1-T442 discovered from *R. oryzae-sativae* TSD190442 showed >90% sequence identity to RosPV1-RS005 discovered in India (Figure 3) [16]. These partitiviruses appeared to be nearly identical to one another despite the vast geographical distance. It is possibly due to the migration of host fungus via international translocation of infected plant materials vice versa, therefore, suggesting their same origin.

### 4.3. Possible Viral Transmission through Hyphal Contact

RosPV2-T442 and RosPV3-T505 from *R. oryzae-sativae* were nearly identical to the partitiviruses of *R. solani* AG-1 IA and *R. fumigata*. This finding leads to the speculation that RosPV2-T442 and RosPV3-T505 could have originated from other basidiomycetous or *Rhizoctonia* host species. Though it has already been proven that *R. solani* can undergo hyphal anastomosis strictly in an intraspecies manner, the same is true for *R. oryzae-sativae* [3,35]. Yet, hyphal contact and cytoplasmic exchange between the two species could have possibly taken place under the field conditions. It is noteworthy that sympatric *R. solani* and *R. oryzae-sativae* were isolated from rice tissues with blighted symptoms in India [36] as well as in this study. Transmission of partitiviruses from donors to taxonomically different recipients is definitely feasible via fungal protoplast transfection with purified virions, which was thoroughly elucidated via in vitro [10,33,37,38,39]. However, there is no evidence that virion introduction could occur under natural environmental conditions. Thus, partitiviruses were presumably migrated across taxonomically incompatible hosts via hyphal fusion. As demonstrated in a laboratory setting, *Heterobasidion abietinum* and *H. parviporum* (of *H. annosum* species complex), which harbored Heterobasidion partitivirus 1 (HQ541323 and HQ541324) and an unassigned partitivirus, Heterobasidion RNA virus 5 strain pa1 (HQ541326), were able to pass on the partitiviruses to *H. ecrustosum* of *H. insulare* species complex via hyphal anastomosis [40]. In the same study, another unassigned partitivirus, Heterobasidion RNA virus 1 strain au1 (HQ541328), infecting *H. austral* of *H. insulare* species complex was also able to infect four different *H. annosum* species. *R. solani*, *R. oryzae-sativae*, and *R. fumigata* belong to the same *R. solani* species complex [41]. RfPV (detected in *R. fumigata)* and RosPV3-T505 (detected in *R. oryzae-sativae)* shared extremely high aa sequence identities (RdRp: 99.4%, CP: 94.5%) despite lingering inside different hosts (Figure 3). Therefore, it is interesting to speculate the possible horizontal transmission of partitiviruses through hyphal contact among these fungi.

### 4.4. Mycovirus-Curing in Rhizoctonia *spp.*

Many isogenic virus-free variants were obtained from their originally virus-infected hosts by simple techniques such as single spore isolation, hyphal tipping, and protoplasting [42,43]. Obtaining virus-cured stain is a crucial step toward the etiological study of the mycoviral impact on fungal host’s transcriptome, small RNAome, proteome, metabolome, lipidome, and epigenome [44]. However, this is often not the case for basidiomycetous fungi, especially in *Rhizoctonia* spp. Many attempts to eliminate mycoviruses from *Rhizoctonia* spp. have been made in the past, but no isogenic sub-isolate could be obtained excepting one case of RsHV2 [27]. Due to such difficulty, possible effects of the virus have been unexplored in many viruses/*Rhizoctonia* spp. combinations (Table 3) [14,15,16,20,26,27,28,29,30,31]. These species do not produce spores, thus limiting the options only to hyphal tipping and protoplasting [20]. Hyphal tips retrieved from the edge of the actively growing mycelia of *R. oryzae-sativae* TSS190442 cultured on PDA with/without ribavirin retained RosPV1 and RosPV2. Likewise, all regenerated protoplasts retained both the partitiviruses, which was confirmed by one-step colony RT-PCR (data not shown). When cycloheximide was applied along with ribavirin, we could eliminate the two partitiviruses. Cycloheximide is a known growth inhibitor that interrupts protein elongation by the ribosomes in the eukaryotic cells, and ribavirin is a nucleoside analog that is integrated into the viral RNA genome and affects the replication of the viral genome [45,46,47]. It has been reported that partitivirus particles contain one or two copies of RdRp molecule to synthesize plus-strand RNA (mRNA) for further translation in the cytoplasm, but its proliferation mainly depends upon the translation by the fungal host ribosome [22]. Thus, it is very tempting to speculate that cycloheximide may have synergistically hampered the process of partitivirus translation and indirectly affected replication. When coupled with the anti-viral drug ribavirin, the transmission of partitiviruses to the newly developed fungal cells might occur inefficiently, producing some fungal cells without the virus. Nonetheless, further study is needed to confirm this hypothesis. The degree of mycovirus adaptation to its host fungus may influence the rate of mycovirus elimination, but this study has proven that hyphal tipping in conjunction with cycloheximide and ribavirin treatment is effective when curing partitiviruses of *R. oryzae-sativae*. This method may open up more opportunities to investigate fungus–mycovirus interaction in other pathosystems as well.

When viruses were eliminated, the virus-free *R. oryzae-sativae* isolate, T442-VF10, showed a slightly improved growth rate and formed a more severe symptom on the rice sheath. This implies that the coinfecting partitiviruses, to some extent, may influence the pathogenicity of their host (Figure 5). The partitivirus infections, in most cases, do not alter the phenotypes (growth, virulence, sporulation, and pigmentation) of their hosts. For example, phenotype and colony morphology of *Trichoderma harzianum* strain NCFC319 remained unaltered when Trichoderma harzianum partitivirus 1 was eliminated from this strain by hyphal tipping [48]. However, instances wherein partitivirus conferred hypervirulent traits or helped to induce hypovirulence in host fungi were also reported [19,33,49]. Nonetheless, further studies are needed to investigate the complex interaction between the two partitiviruses, RosPV1 and RosPV2, and their *R. oryzae-sativae* host.

In addition, *R. solani* TSD190117, possessing dsRNA elements that are similar in size with partitiviruses, exhibited extremely impaired growth on synthetic media (Appendix A). However, it is unknown whether the phenotype is associated with mycoviral infection. Nonetheless, it would be of great interest to apply the developed virus-curing protocol for understanding the phenomenon.

## Figures and Tables

**Figure 1 viruses-13-02269-f001:**
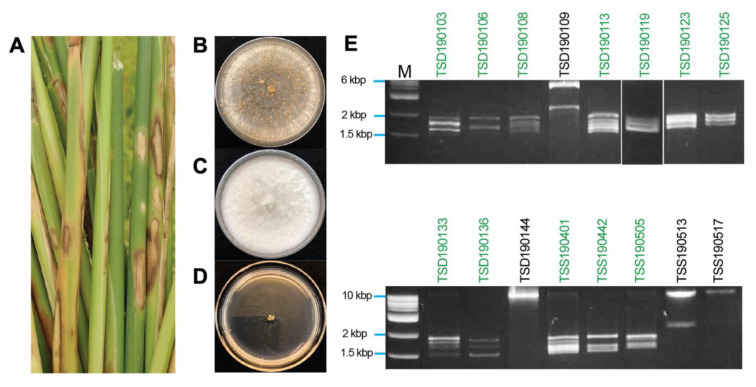
Isolation of rice pathogenic fungi and mycovirus dsRNA profiles. (**A**) Rice sheaths with the blighted symptoms. Diseased leaf tissues were collected from paddies in Thailand and further subjected to single fungal species isolation. (**B**–**D**) Colony morphology of isolated fungi including *Rhizoctonia oryzae-sativae* (TSS190505 isolate) (**B**), *Nigrospora oryzae* (TSD190109 isolate) (**C**), and *Rhizoctonia solani* (TSD190117 isolate) (**D**). (**E**) Electrophoretic profile of dsRNAs purified from fungal isolates. The majority of the dsRNA elements appeared to be within the size range of approximately 1.5 to 2 kbp. Green letters above the panels indicate *R. oryzae-sativae* isolates, while those in black letters are others.

**Figure 2 viruses-13-02269-f002:**
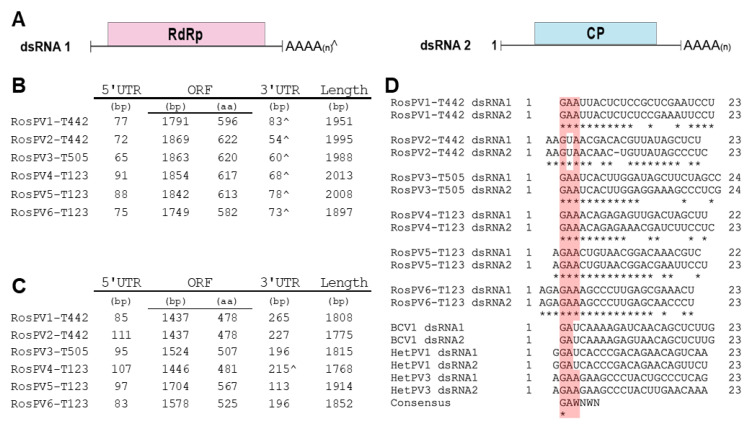
Schematic representation of the genomic composition of the six partitiviruses discovered from three *R. oryzae-sativae* isolates. (**A**) Genomic organization of the two dsRNAs that constitute a single partitivirus. The red and blue rectangular boxes represent open reading frames (ORFs) in each segment encoding RdRp and CP. (**B**,**C**) The genomic composition of the dsRNA1 and dsRNA2 of each partitivirus are described in B, and C, respectively. ^ indicates interrupted poly(A) tails at 3′ UTR. (**D**) A portion of 5′ UTR sequences was aligned with previously-characterized alphapartitiviruses showing conserved residues at the terminal regions. The pink-highlighted box illustrates the conserved residues in the aligned sequences. Consensus was obtained from [22] to further support the conservation of the plus strand terminal regions of alphapartitiviruses.

**Figure 3 viruses-13-02269-f003:**
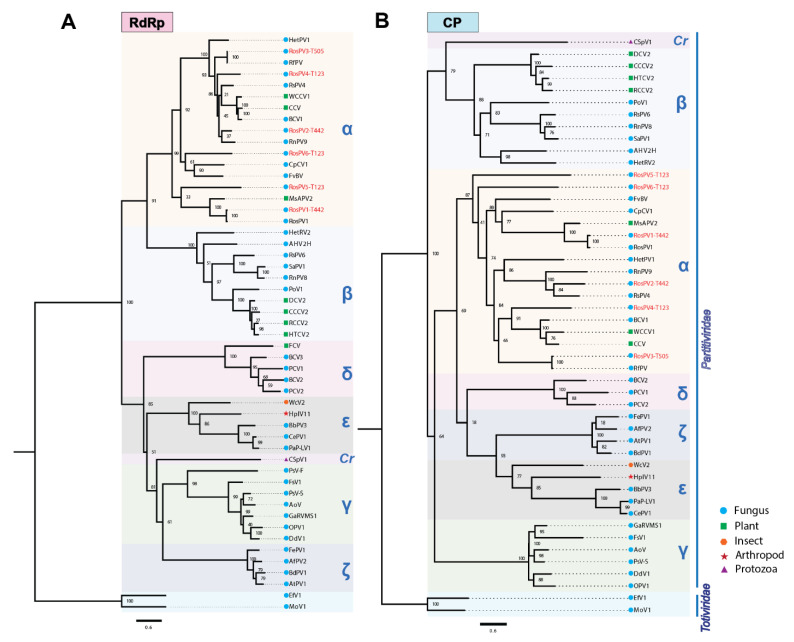
Phylogenetic analyses based on the deduced aa sequences from the six partitiviruses. (**A**,**B**) Maximum-likelihood trees were constructed based on the alignment of complete aa sequences of RdRp (**A**) and CP (**B**) along with the previously reported members of the genera or proposed genera of the family *Partitividae*. Two members of the family *Totiviridae* were used as an outgroup for rooting. Tree construction parameters were chosen based on Smart Model Selection (SMS) function available in PhyML (http://www.atgc-montpellier.fr/sms/, accessed on 3 November 2021). Blanch support was obtained with the aLRT-SH-like method and indicated beside each node in percentage (1.00 = 100%). Red texts indicate partitiviruses discovered in this study. The scale bars represent genetic distance. α, *Alphapartitivirus*; β, *Betapartitivirus*; γ, *Gammapartitivirus*; δ, *Deltapartitivirus*; *Cr*, *Cryspovirus*; ε, “Epsilonpartitivirus”; and ζ, “Zetapartitivirus”. Accession number and the full name of viruses used in this phylogenetic tree constructions can be found in Appendix A.

**Figure 4 viruses-13-02269-f004:**
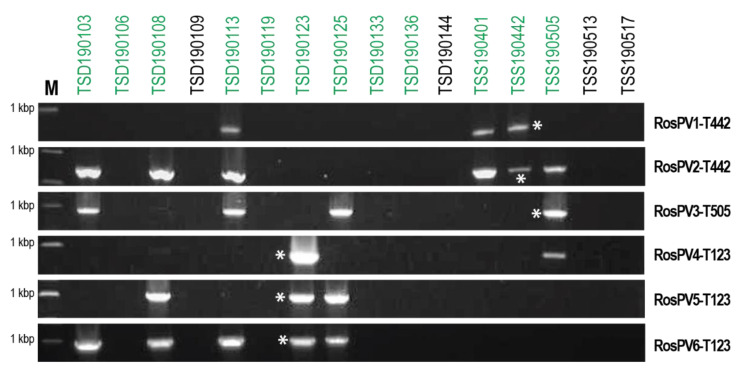
Widespread infection of partitiviruses among *R. oryzae-sativae* isolates. cDNAs synthesized from total RNA fractions were used as templates to detect the presence of the six partitiviruses, RosPV1 to RosPV6 (indicated on the right side of panels). Coinfection of more than two partitiviruses in a single isolate was observed but no single infection. M indicates a DNA size marker in the length of 1 kbp, and green letters above the panel indicate *R. oryzae-sativae* isolates. Asterisks represent positive control by which each partitivirus was detected from their original host.

**Figure 5 viruses-13-02269-f005:**
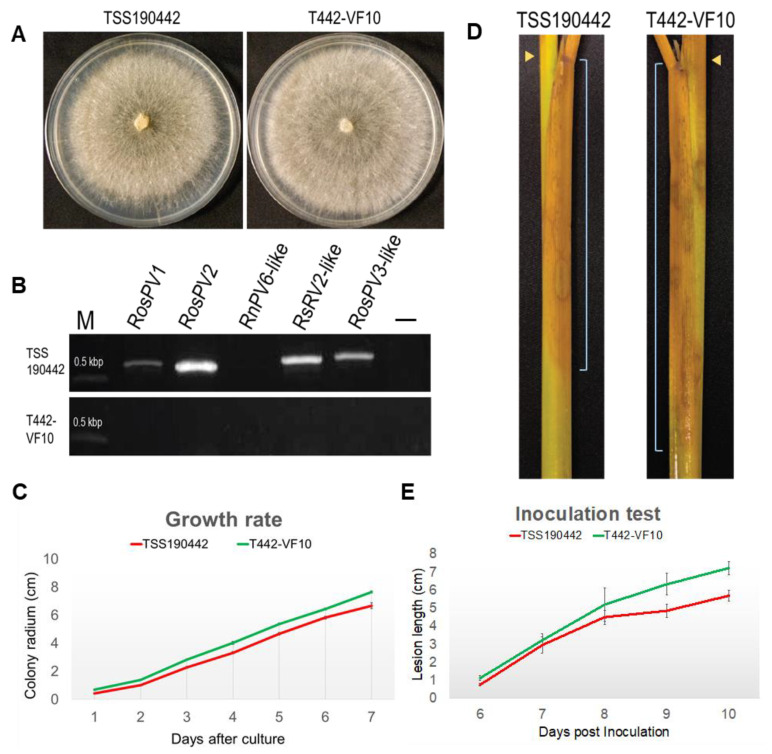
Biological impact of *partitivirus* infections in *R. oryzae-sativae* TSS190442. Isogenic virus-free *R. oryzae-sativae* isolate T442-VF10 was successfully obtained from RosPV1- and RosPV2-infected TSS190442 by the successive treatments of ribavirin and cycloheximide. (**A**) Colony morphological comparison between virus-infected and virus-free isolates. (**B**) RT-PCR detection of two fully and three partially characterized partitiviruses in TSS190442 and T442-VF10. (**C**) Growth of fungi on the synthetic media. Mycelial plugs were inoculated at the edge of PDA plates and the radius of the colony were measured. (**D**,**E**) Symptom development on rice tissues. Mycelial plugs were inoculated at the node of rice sheath (**D**, yellow arrowheads) and lesions were observed daily-basis. The symptom appeared downward. The distance from the inoculation point to the edge of the symptom (scale bars in **D**) was measured (**E**). The virus-cured isolate T442-VF10 showed a slightly faster growth rate on both the plate and plant; however, this significance was not statistically supported (**C**,**E**).

**Table 1 viruses-13-02269-t001:** List of fungal species roughly predicted by BLASTn analysis of ITS sequences, and presence of dsRNA in each possible species.

Fungus Species	Number of Isolates	dsRNA (%) *	Fungus Species	Number of Isolates	dsRNA (%) *
*Achroiostachys saccharicxola*	3	0 (0%)	*Nigrospora sphaerica*	2	1 (50%)
*Bipolaris oryzae*	1	0 (0%)	*Sarocladium oryzae*	1	0 (0%)
*Bipolaris sivanesaniana*	7	0 (0%)	*Sclerotium hydrophilum*	2	0 (0%)
*Curvularia beasleyi*	1	0 (0%)	*Setosphaeria rostrata*	2	0 (0%)
*Fusarium equiseti*	1	0 (0%)	*Simplicillium lamellicola*	1	0 (0%)
*Fusarium incarnatum*	1	0 (0%)	*Stachybotrys* cf. *elegans*	2	0 (0%)
*Fusarium proliferatum*	4	0 (0%)	*Rhizoctonia oryzae*	3	1 (33%)
*Fusarium sacchari*	1	0 (0%)	*Rhizoctonia oryzae-sativae*	37	12 (36%)
*Gaeumannomyces oryzinus*	5	0 (0%)	*Rhizoctonia solani*	1	1 (100%)
*Nigrospora oryzae*	4	1 (25%)	*Rhizoctonia zeae*	1	1 (100%)

* Percentage of dsRNA-positive isolates belonging to a particular fungal species.

**Table 2 viruses-13-02269-t002:** Sequence identities of cDNA contigs to newly or previously reported partitiviruses’ protein (RdRp or CP) by BLASTx.

*Rhizoctonia oryzae-sativae* Isolates		The aa Sequence Identities to Reported Partitiviruses (%)											
		RosPV1-T442 ^+^	RosPV2-T442 ^+^	RosPV3-T505 ^+^	RosPV4-T123 ^+^	RosPV5-T123 ^+^	RosPV6-T123 ^+^	RsRV2 *	RsRV3 *	RsRV4 *	RsRV5 *	RnPV6 *	RnPV16 *
TSD190103	RdRp		99%	99%									
	CP												
TSD190108	RdRp		99%	91%						86%			
	CP		84%										
TSD190123	RdRp				100%	100%	100%						
	CP				100%	100%	100%						
TSS190401	RdRp		91%								99%		92%
	CP	83%											
TSS190442	RdRp	100%	100%	92%									
	CP	100%	100%					88%				86%	
TSS190505	RdRp		94%	100%									
	CP			100%					15%				

+ Represents partitiviruses discovered in this study and is shown as 100% aa identity in the column. * Previously reported mycoviruses, Rhizoctonia solani dsRNA virus 2–5 (RsRV2–5), Rosellinia necatrix partitivirus 6 and 16 (RnPV6 and 16).

**Table 3 viruses-13-02269-t003:** An overview of mycovirus elimination in *Rhizoctonia* spp.

Mycovirus	*Rhizoctonia* spp.		Virus-Curing Attempt	Method	Result of Virus-Curing	Source
	Species	Strain				
RsRV3	*R. solani*	A105	−	N/A	N/A	[28]
RsPV5	*R. solani*	C24	−	N/A	N/A	[26]
RsRV1	*R. solani*	B275	−	N/A	N/A	[29]
RsRV-HN008	*R. solani*	HN008	−	N/A	N/A	[30]
RosPV1	*R. oryzae-sativae*	RS005	−	N/A	N/A	[16]
RoMV1	*R. oryzae-sativae*	89-1	−	N/A	N/A	[15]
RsEV1	*R. solani*	GD-2	+	Hyphal tipping	Unsuccessful	[14]
RsPV2	*R. solani*	GD-11	+	Hyphal tipping	Unsuccessful	[20]
RsPV6, 7, 8	*R. solani*	YNBB-111	+	Hyphal tipping and ribavirin	Unsuccessful	[31]
				Protoplasting	Unsuccessful	
RsHV2	*R. solani*	XN84	+	Protoplasting	Successful	[27]
RosPV1 & 2-T442	*R. oryzae-sativae*	TSD190442	+	Protoplasting	Unsuccessful	
				Heat treatment	Unsuccessful	
				Hyphal tipping	Unsuccessful	
				Hyphal tipping and ribavirin	Unsuccessful	This study
				Hyphal tipping and ribavirin and cycloheximide	Successful	

−/+ Attempt to cure mycovirus. N/A: Not applicable. RsHV2: Rhizoctonia solani hypovirus 2.

## Data Availability

The complete viral cDNA sequences were deposited under the GenBank accession number LC637850–LC637861. Since the RNA-seq data (DRA) includes unpublished items, it will be publicly opened once the remaining data being reported elsewhere.

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
