# Peer review of "Omnipresence of Partitiviruses in Rice Aggregate Sheath Spot Symptom-Associated Fungal Isolates from Paddies in Thailand"

_viruses, 2021, doi:10.3390/v13112269_

Round 1
Reviewer 1 Report
Line 23, ‘ranged’ is changed to ‘ranging’
Line 71, ‘Deltapartitivirus’ is changed to ‘Deltapartitivirus,’
Line 77, ‘Beta-’ is changed to ‘Beta-,’
Line 101, ‘five days’ are replaced by ‘5 d’
Line 122, please provide the full name of STE
Line 126, please provide the full name of TBE
Line 148 and Line 160, ‘Takara’ is changed to ‘TaKaRa’
Line 166, ‘cDNA library construction’ are changed to ‘cDNA library construction,’
Line 187, ‘Blast’ is replaced by ‘BLAST’
Line 238, ‘seven days’ are replaced by ‘7 d’
Line 259 and Line 266, ‘Blastn’ is replaced by ‘BLASTn’
In Table 1, ‘isolates’ are replaced by ‘Number of isolates’
Line 288, please provide the AG or subgroup for the and Rhizoctonia solani isolate TSD190117
Line 302, ‘Genbank’ is changed to ‘GenBank’
Line 306, ‘2008’ is changed to ‘2008,’
Line 307, ‘43.4%’ is changed to ‘43.4%,’
Line 315, ‘were ranged’ are changed to ‘ranged’
Line 327, ‘Alphapartitivirus’ is changed to ‘the genus Alphapartitivirus’
Line 336, Line 337, Line 367, Line 415, and Line 492, ‘Blast’ is replaced by ‘BLAST’
Line 377, ‘family’ is changed to ‘the family’
Line 438, ‘were reported unsuccessful’ are changed to ‘were reported to be unsuccessful’
Line 447, ‘positive’ is changed to ‘to be positive’
Line 464, ‘RosPV2’ is changed to ‘RosPV2,’
Line 472, ‘Partitivirus’ is changed to ‘partitivirus’
Line 476, ‘measured the radius of the colony’ are changed to ‘the radius of the colony were measured’
Line 521, ‘Partitiviridae family’ are replaced by ‘family Partitiviridae’
Line 529, ‘in China’ are changed to ‘in China;’
Line 637, ‘Efficacy of Natural Plant Products on the Control of Aggregate Sheath Spot of Rice’ are changed to ‘Efficacy of natural plant products on the control of aggregate sheath spot of rice’
Line 639-640, ‘Ceratobasidium oryzae-sativae sp. nov., the Teleomorph of Rhizoctonia oryzae-sativae and Ceratobasidium setariae comb. nov., the Probable Teleomorph’ are changed to ‘Ceratobasidium oryzae-sativae sp. nov., the teleomorph of Rhizoctonia oryzae-sativae and Ceratobasidium setariae comb. nov., the probable teleomorph’; In addition, ‘Ceratobasidium oryzae-sativae and Ceratobasidium setariae’ should be italicized.
Line 644-645, ‘Mycoviruses in Biological Control: From Basic Research to Field Implementation’ are changed to ‘Mycoviruses in biological control: from basic research to field implementation’
Line 649-650, ‘Infectious cDNA Clone of Hypovirus CHV1-Euro7: a Comparative Virology Approach To Investigate Virus-Mediated Hypovirulence of the Chestnut Blight Fungus’ are changed to ‘Infectious cDNA clone of hypovirus CHV1-Euro7: a comparative virology approach to investigate virus-mediated hypovirulence of the chestnut blight fungus’
Line 654-656, ‘Novel Bipartite Double-Stranded RNA Mycovirus from the White Root Rot Fungus Rosellinia necatrix: Molecular and Biological Characterization, Taxonomic Considerations, and Potential for Biological Control’ are changed to ‘Novel bipartite double-stranded RNA mycovirus from the white root rot fungus Rosellinia necatrix: molecular and biological characterization, taxonomic considerations, and potential for biological control’
Line 657-658, ‘A Novel Partitivirus That Confers Hypovirulence on Plant Pathogenic Fungi’ are changed to ‘A novel partitivirus that confers hypovirulence on plant pathogenic fungi’
Line 669, ‘rhizoctonia solani ag-1 ia’ are changed to ‘Rhizoctonia solani AG-1 IA’, and Rhizoctonia solani should be italicized.
Line 672, ‘Rhizoctonia oryzae-sativae’ should be italicized.
Line 678-679, ‘A Novel Partitivirus in the Hypovirulent Isolate QT5-19 of the Plant Pathogenic Fungus’ are changed to ‘A novel partitivirus in the hypovirulent isolate QT5-19 of the plant pathogenic fungus’
Line 681, ‘Rhizoctonia solani’ should be italicized.
Line 688, ‘Diverse Partitiviruses From the Phytopathogenic Fungus’ are changed to ‘Diverse partitiviruses from the phytopathogenic fungus’
Line 692-693, ‘Extreme Diversity of Mycoviruses Present in Isolates of Rhizoctonia solani AG2-2 LP From Zoysia japonica From’ are changed to ‘Extreme diversity of mycoviruses present in isolates of Rhizoctonia solani AG2-2 LP from Zoysia japonica from’; in addition, Zoysia japonica should be italicized.
Line 695-696, ‘Complete Nucleotide Sequence of a Partitivirus from Rhizoctonia solani AG-1 IA Strain C24’ are changed to ‘Complete nucleotide sequence of a partitivirus from Rhizoctonia solani AG-1 IA strain C24’.
Line 711-712, ‘Extending the Fungal Host Range of a Partitivirus and a Mycoreovirus from Rosellinia necatrix by Inoculation of Protoplasts with Virus Particles.’ are changed to ‘Extending the fungal host range of a partitivirus and a mycoreovirus from Rosellinia necatrix by inoculation of protoplasts with virus particles.’; in addition, Rosellinia necatrix should be italicized.
Line 714, ‘Population Biology of the Rhizoctonia solani Complex’ are changed to ‘Population biology of the Rhizoctonia solani complex’.
Line 718-719, ‘Characterization, Genetic Structure, and Pathogenicity of Rhizoctonia spp. Associated with Rice Sheath Diseases’ are changed to ‘Characterization, genetic structure, and pathogenicity of Rhizoctonia spp. associated with rice sheath diseases’
Line 720-721, ‘A Novel Victorivirus from a Phytopathogenic Fungus, Rosellinia necatrix, Is Infectious as Particles and Targeted by RNA Silencing’ are changed to ‘A novel victorivirus from a phytopathogenic fungus, Rosellinia necatrix, is infectious as particles and targeted by RNA silencing’; in addition, Rosellinia necatrix should be italicized.
Line 722-723, ‘Effects of Defective Interfering RNA on Symptom Induction by, and Replication of, a Novel Partitivirus from a Phytopathogenic Fungus’ are changed to ‘Effects of defective interfering RNA on symptom induction by and replication of, a novel partitivirus from a phytopathogenic fungus’
Line 730-731, ‘Characterization of Anastomosis Groups of Binucleate Rhizoctonia Species Using Restriction Analysis of an Amplified Ribosomal RNA Gene’ are changed to ‘Characterization of anastomosis groups of binucleate Rhizoctonia species using restriction analysis of an amplified ribosomal RNA gene’
Line 732, ‘Sclerotium rolfsii’ should be italicized.
Line 739, ‘Ustilaginoidea virens’ should be italicized.
Line 743, ‘Identification of a Novel Partitivirus of Trichoderma harzianum NFCF319 and Evidence for the Related Antifungal Activity’ are changed to ‘Identification of a novel partitivirus of Trichoderma harzianum NFCF319 and evidence for the related antifungal activity’
Author Response
Thank you very much for many corrections. We have modified the manuscript based on the comments provided by the reviewer. All modifications in the main text were highlighted with yellow color, and answer to the comments, suggestions, and questions are as below:
Line 23, ‘ranged’ is changed to ‘ranging’
>the word was corrected accordingly (L22)
Line 71, ‘Deltapartitivirus’ is changed to ‘Deltapartitivirus,’
>the word was corrected accordingly (L70)
Line 77, ‘Beta-’ is changed to ‘Beta-,’
>the word was corrected accordingly (L76)
Line 101, ‘five days’ are replaced by ‘5 d’
>the word was corrected accordingly (L100)
Line 122, please provide the full name of STE
>the part was modified into ”1x Sodium/Tris/EDTA (100 mM NaCl, 10 mM Tris-HCl pH 8.0, 1 mM EDT pH 8.0)” (L121)
Line 126, please provide the full name of TBE
>the part was modified as ”0.5x Tris/Borate/EDTA (40 mM Tris, 45 mM boric acid, 1 mM EDTA)” (L125)
Line 148 and Line 160, ‘Takara’ is changed to ‘TaKaRa’
>the word was corrected accordingly (L148, L160)
Line 166, ‘cDNA library construction’ are changed to ‘cDNA library construction,’
>the word was corrected accordingly (L166)
Line 187, ‘Blast’ is replaced by ‘BLAST’
>the word was corrected accordingly (L187)
Line 238, ‘seven days’ are replaced by ‘7 d’
>the word was corrected accordingly (L238)
Line 259 and Line 266, ‘Blastn’ is replaced by ‘BLASTn’
>the word was corrected accordingly (L259, L265)
In Table 1, ‘isolates’ are replaced by ‘Number of isolates’
>the word was corrected accordingly (Table 1)
Line 288, please provide the AG or subgroup for the and Rhizoctonia solani isolate TSD190117
>the fungal isolate of this highest BLASTn hit with the ITS-amplicon of TSD190117 as a query was R. solani clone RS3 (MK213723), but no information about subgrouping of the isolate was reported (Suryawanshi Padmaja Pralha, P.U. Krishnaraj and S.K. Prashanthi, 2019, Int.J.Curr.Microbiol.App.Sci (2019) 8(1): 1714-1721)
Line 302, ‘Genbank’ is changed to ‘GenBank’
>the word was corrected accordingly (L301)
Line 306, ‘2008’ is changed to ‘2008,’
>the word was corrected accordingly (L305)
Line 307, ‘43.4%’ is changed to ‘43.4%,’
>the word was corrected accordingly (L306)
Line 315, ‘were ranged’ are changed to ‘ranged’
>the word was corrected accordingly (L314)
Line 327, ‘Alphapartitivirus’ is changed to ‘the genus Alphapartitivirus’
>the word was corrected accordingly (L326)
Line 336, Line 337, Line 367, Line 415, and Line 492, ‘Blast’ is replaced by ‘BLAST’
>the word was corrected accordingly (L335, 336, 414, 524)
Line 377, ‘family’ is changed to ‘the family’
>the word was corrected accordingly (L375)
Line 438, ‘were reported unsuccessful’ are changed to ‘were reported to be unsuccessful’
>the word was corrected accordingly (L440)
Line 447, ‘positive’ is changed to ‘to be positive’
>the word was corrected accordingly (L449)
Line 464, ‘RosPV2’ is changed to ‘RosPV2,’
>the word was corrected accordingly (L466)
Line 472, ‘Partitivirus’ is changed to ‘partitivirus’
>the word was corrected accordingly (474)
Line 476, ‘measured the radius of the colony’ are changed to ‘the radius of the colony were measured’
>the word was corrected accordingly (L478)
Line 521, ‘Partitiviridae family’ are replaced by ‘family Partitiviridae’
>the word was corrected accordingly (L553)
Line 529, ‘in China’ are changed to ‘in China;’
>the word was corrected accordingly (L561)
Line 637, ‘Efficacy of Natural Plant Products on the Control of Aggregate Sheath Spot of Rice’ are changed to ‘Efficacy of natural plant products on the control of aggregate sheath spot of rice’
>the word was corrected accordingly (L670)
Line 639-640, ‘Ceratobasidium oryzae-sativae sp. nov., the Teleomorph of Rhizoctonia oryzae-sativae and Ceratobasidium setariae comb. nov., the Probable Teleomorph’ are changed to ‘Ceratobasidium oryzae-sativae sp. nov., the teleomorph of Rhizoctonia oryzae-sativae and Ceratobasidium setariae comb. nov., the probable teleomorph’; In addition, ‘Ceratobasidium oryzae-sativae and Ceratobasidium setariae’ should be italicized.
>the word was corrected accordingly (L672)
Line 644-645, ‘Mycoviruses in Biological Control: From Basic Research to Field Implementation’ are changed to ‘Mycoviruses in biological control: from basic research to field implementation’
>the word was corrected accordingly (L677)
Line 649-650, ‘Infectious cDNA Clone of Hypovirus CHV1-Euro7: a Comparative Virology Approach To Investigate Virus-Mediated Hypovirulence of the Chestnut Blight Fungus’ are changed to ‘Infectious cDNA clone of hypovirus CHV1-Euro7: a comparative virology approach to investigate virus-mediated hypovirulence of the chestnut blight fungus’
>the word was corrected accordingly (L682)
Line 654-656, ‘Novel Bipartite Double-Stranded RNA Mycovirus from the White Root Rot Fungus Rosellinia necatrix: Molecular and Biological Characterization, Taxonomic Considerations, and Potential for Biological Control’ are changed to ‘Novel bipartite double-stranded RNA mycovirus from the white root rot fungus Rosellinia necatrix: molecular and biological characterization, taxonomic considerations, and potential for biological control’
>the word was corrected accordingly (L687)
Line 657-658, ‘A Novel Partitivirus That Confers Hypovirulence on Plant Pathogenic Fungi’ are changed to ‘A novel partitivirus that confers hypovirulence on plant pathogenic fungi’
>the word was corrected accordingly (L690)
Line 669, ‘rhizoctonia solani ag-1 ia’ are changed to ‘Rhizoctonia solani AG-1 IA’, and Rhizoctonia solani should be italicized.
>the word was corrected accordingly (L702)
Line 672, ‘Rhizoctonia oryzae-sativae’ should be italicized.
>the word was corrected accordingly (L705)
Line 678-679, ‘A Novel Partitivirus in the Hypovirulent Isolate QT5-19 of the Plant Pathogenic Fungus’ are changed to ‘A novel partitivirus in the hypovirulent isolate QT5-19 of the plant pathogenic fungus’
>the word was corrected accordingly (L711)
Line 681, ‘Rhizoctonia solani’ should be italicized.
>the word was corrected accordingly (L714)
Line 688, ‘Diverse Partitiviruses From the Phytopathogenic Fungus’ are changed to ‘Diverse partitiviruses from the phytopathogenic fungus’
>the word was corrected accordingly (L721)
Line 692-693, ‘Extreme Diversity of Mycoviruses Present in Isolates of Rhizoctonia solani AG2-2 LP From Zoysia japonica From’ are changed to ‘Extreme diversity of mycoviruses present in isolates of Rhizoctonia solani AG2-2 LP from Zoysia japonica from’; in addition, Zoysia japonica should be italicized.
>the word was corrected accordingly (L725)
Line 695-696, ‘Complete Nucleotide Sequence of a Partitivirus from Rhizoctonia solani AG-1 IA Strain C24’ are changed to ‘Complete nucleotide sequence of a partitivirus from Rhizoctonia solani AG-1 IA strain C24’.
>the word was corrected accordingly (L728)
Line 711-712, ‘Extending the Fungal Host Range of a Partitivirus and a Mycoreovirus from Rosellinia necatrix by Inoculation of Protoplasts with Virus Particles.’ are changed to ‘Extending the fungal host range of a partitivirus and a mycoreovirus from Rosellinia necatrix by inoculation of protoplasts with virus particles.’; in addition, Rosellinia necatrix should be italicized.
>the word was corrected accordingly (L744)
Line 714, ‘Population Biology of the Rhizoctonia solani Complex’ are changed to ‘Population biology of the Rhizoctonia solani complex’.
>the word was corrected accordingly (L747)
Line 718-719, ‘Characterization, Genetic Structure, and Pathogenicity of Rhizoctonia spp. Associated with Rice Sheath Diseases’ are changed to ‘Characterization, genetic structure, and pathogenicity of Rhizoctonia spp. associated with rice sheath diseases’
>the word was corrected accordingly (L751)
Line 720-721, ‘A Novel Victorivirus from a Phytopathogenic Fungus, Rosellinia necatrix, Is Infectious as Particles and Targeted by RNA Silencing’ are changed to ‘A novel victorivirus from a phytopathogenic fungus, Rosellinia necatrix, is infectious as particles and targeted by RNA silencing’; in addition, Rosellinia necatrix should be italicized.
>the word was corrected accordingly (L753)
Line 722-723, ‘Effects of Defective Interfering RNA on Symptom Induction by, and Replication of, a Novel Partitivirus from a Phytopathogenic Fungus’ are changed to ‘Effects of defective interfering RNA on symptom induction by and replication of, a novel partitivirus from a phytopathogenic fungus’
>the word was corrected accordingly (L755)
Line 730-731, ‘Characterization of Anastomosis Groups of Binucleate Rhizoctonia Species Using Restriction Analysis of an Amplified Ribosomal RNA Gene’ are changed to ‘Characterization of anastomosis groups of binucleate Rhizoctonia species using restriction analysis of an amplified ribosomal RNA gene’
>the word was corrected accordingly (L763)
Line 732, ‘Sclerotium rolfsii’ should be italicized.
>the word was corrected accordingly (L765)
Line 739, ‘Ustilaginoidea virens’ should be italicized.
>the word was corrected accordingly (L772)
Line 743, ‘Identification of a Novel Partitivirus of Trichoderma harzianum NFCF319 and Evidence for the Related Antifungal Activity’ are changed to ‘Identification of a novel partitivirus of Trichoderma harzianum NFCF319 and evidence for the related antifungal activity’
>the word was corrected accordingly (L776)
Reviewer 2 Report
A few suggestions for revision:
- the authors used "strains" and "isolates" in the manuscript. I suggest using "isolates" in the whole manuscript.
- In Table 1, "Blastn" should be chnaged to "BLASTn"
- In dsRNA screening, the authors showed that many fungi did not have dsRNAs. This might not be true, as the authors did not use RNA-seq technique. They used only dsRNA extraction technique. The conclusion should be very careful.
- Figures 1, 2 and tables are too wide in size, They should be reformated.
- The word "spp." should not be written in italic
- The authors obtained virus-free isogenic isolates, why they were not used as recipients to be re-infected by mycoviruses. The re-introduction of a certain partitivirs may help to understand its role in mediation of the host pathogencity (virulence).
- More parttitivurues should be included in the phylogenitic trees. Host information should be added in the phylogenetic trees
Author Response
Thank you very much for the suggestions. We have modified the manuscript based on the comments provided by the reviewer. All modifications in the main text were highlighted with yellow color, and answer to the comments, suggestions, and questions are as below:
- the authors used “strains” and “isolates” in the manuscript. I suggest using “isolates” in the whole manuscript.
>we have changed most of “strains” indicating Rhizoctonia oryzae-sativae to “isolates” throughout the manuscript, but retained “strains” referring ones previously established
- In Table 1, “Blastn” should be changed to “BLASTn”
>the term was modified accordingly (L265)
- In dsRNA screening, the authors showed that many fungi did not have dsRNAs. This might not be true, as the authors did not use RNA-seq technique. They used only dsRNA extraction technique. The conclusion should be very careful.
>possibility of mycovirus infection that is below the detection limit was additionally mentioned (L518)
- Figures 1, 2 and tables are too wide in size, they should be reformated.
>all figures and tables were resized as suggested
- The word “spp.” should not be written in italic
>the word was corrected except for those in the section title
- The authors obtained virus-free isogenic isolates, why they were not used as recipients to be re-infected by mycoviruses. The re-introduction of a certain partitivirus may help to understand its role in the mediation of the host pathogenicity (virulence).
>re-entry of three partitiviruses are underway, and we think it is possible to obtain re-infected R.os isolates. However, we currently cannot have well-grown rice plants due to the cold season in Japan. Since we have other sets of uncharacterized mycoviruses in the same sources, we would like to address this experiment in the coming year.
- More partitivurues should be included in the phylogenetic trees. Host information should be added in the phylogenetic trees
>thanks to the suggestion, we have a new figure of improved trees accomodating more partitiviruses and host information (Fig. 3).
Reviewer 3 Report
The authors have screened for dsRNA mycoviruses in Thai isolates of rice fungal pathogen Rhizoctonia oryzae-sativae, the causal agent of rice sheath spot disease and identified six new alphapartitiviruses. The authors have also successfully cured the fungi of the viruses and showed that the cured fungi increases the severity of the symptoms.
The design of the experiments, analysis and interpretation of the results and preparation of the manuscript are very good.
Minor comment:
Why did the authors specifically look only for dsRNA viruses? It was also highlighted by the authors in few contexts that there could be additional ssRNA/dsRNA viruses in the fungi. The RNA used for the analysis is treated with DNases and S1 nuclease. Although less efficiently, the S1 nuclease can also digest the ssRNA and this could have eliminated identification of possible ssRNA viruses. Can the authors please comment on this?
Could it be possible that the first adaptor-tagged-random hexamer primer used for the synthesis of the first-strand cDNA synthesis could have contributed to the bias in cloning and some dsRNA viruses were not cloned? Could a more holistic RNAseq analysis have identified all the mycoviruses in the fungal samples?
Author Response
Thank you very much for comments and questions. We have modified the manuscript based on the below communication with the reviewer. All modifications in the main text were highlighted with yellow color, and answer to the comments, suggestions, and questions are as below:
Why did the authors specifically look only for dsRNA viruses? It was also highlighted by the authors in few contexts that there could be additional ssRNA/dsRNA viruses in the fungi. The RNA used for the analysis is treated with DNases and S1 nuclease. Although less efficiently, the S1 nuclease can also digest the ssRNA and this could have eliminated the identification of possible ssRNA viruses. Can the authors please comment on this?
>We focused on “visible” dsRNA elements (expected viral origin) that were ~ 2 kbp in size and were commonly detected in R. oryzae-sativae, and this is why S1 nuclease was used to eliminate potential contaminants of ssRNAs. As a result, we found those dsRNA elements are mainly of dsRNA genome segments of alphapartitiviruses and discovered the omnipresent nature of these viruses in the host fungi in the paddies. Besides, of course, ssRNA viruses may be coinfecting the same host isolates used in this study. Indeed, we found many ssRNA viruses in the isolates subjected to the RNA-seq analysis (R. oryzae-sativae TSD 190123). But again, these viruses are not in the scope of this study but partitiviruses. A few sentences from this viewpoint were added to make things more clear (L173, L530~L534).
Could it be possible that the first adaptor-tagged-random hexamer primer used for the synthesis of the first-strand cDNA synthesis could have contributed to the bias in cloning and some dsRNA viruses were not cloned? Could a more holistic RNAseq analysis have identified all the mycoviruses in the fungal samples?
>Other mycoviruses might present as mentioned above. The bias in cDNA library construction could be one of the reasons for the unsuccessful detection of such hidden viruses. We think this potential bias was enhanced by the different accumulation levels of dsRNA templates between partitiviral dsRNAs and others dsRNA elements including replicative intermediates of ssRNA viruses and dsRNA viruses with a low multiplication capacity.
On the other hand, the RNA-seq approach (rRNA-depleted RNA samples) was considered highly sensitive to detecting these non-partitiviruses in TSD190123. These viruses will be reported elsewhere together with viruses found in Nigrospora species.